# Active 3D Shape Reconstruction
# from Vision and Touch

**Edward J. Smith**[1,2]*     **David Meger**[2]     **Luis Pineda**[1]     **Roberto Calandra**[1]

**Jitendra Malik**[1,3]     **Adriana Romero-Soriano**[1,2,†]     **Michal Drozdzal**[1,†]

[1] Facebook AI Research    [2] McGill University    [3] University of California, Berkeley

## Abstract

Humans build 3D understandings of the world through *active object exploration*, using jointly their senses of vision and touch. However, in 3D shape reconstruction, most recent progress has relied on *static datasets* of limited sensory data such as RGB images, depth maps or haptic readings, leaving the active exploration of the shape largely unexplored. In *active touch sensing* for 3D reconstruction, the goal is to actively select the tactile readings that maximize the improvement in shape reconstruction accuracy. However, the development of deep learning-based active touch models is largely limited by the lack of frameworks for shape exploration. In this paper, we focus on this problem and introduce a system composed of: 1) a haptic simulator leveraging high spatial resolution vision-based tactile sensors for active touching of 3D objects; 2) a mesh-based 3D shape reconstruction model that relies on tactile or visuotactile signals; and 3) a set of data-driven solutions with either tactile or visuotactile priors to guide the shape exploration. Our framework enables the development of the first fully data-driven solutions to active touch on top of learned models for object understanding. Our experiments show the benefits of such solutions in the task of 3D shape understanding where our models consistently outperform natural baselines. We provide our framework as a tool to foster future research in this direction.

## 1 Introduction

3D shape understanding is an active area of research, whose goal is to build 3D models of objects and environments from limited sensory data. It is commonly tackled by leveraging partial observations such as a single view RGB image [60, 67, 23, 44], multiple view RGB images [16, 25, 30, 31], depth maps [61, 74] or tactile readings [7, 49, 59, 45, 38]. Most of this research focuses on building shape reconstruction models from a *fixed set* of partial observations. However, this constraint is relaxed in the *active sensing* scenario, where additional observation can be acquired to improve the quality of the 3D reconstructions. In *active vision* [4], for instance, the objective can be to iteratively select camera perspectives from an object that result in the highest improvement in quality of the reconstruction [76] and only very recently the research community has started to leverage large scale datasets to learn exploration strategies that generalize to unseen objects [77, 5, 41, 51, 29, 78, 50, 6].

Human haptic exploration of objects both with and without the presence of vision has classically been analysed from a psychological perspective, where it was discovered that the developed tactile exploration strategies for object understanding were demonstrated to not only be ubiquitous, but also highly tailored to specific tasks [37, 33]. In spite of this, deep learning-based data-driven approaches to *active touch* for shape understanding are practically non-existent. Previous haptic exploration works consider objects

---

*Correspondence to: ejsmith@fb.com and edward.smith@mail.mcgill.ca
†Equal Contribution

35th Conference on Neural Information Processing Systems (NeurIPS 2021), virtual.

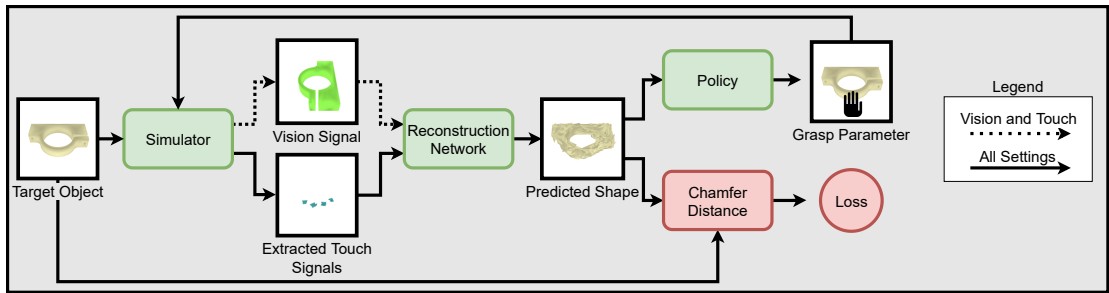

Figure 1: An overview of our active touch exploration framework. Given a 3D model, the simulator extracts touch and vision signals that are fed to the reconstruction model. The reconstruction model predicts a 3D shape that is used as an input to a policy model that decides where to touch next. The policies are trained to select grasps which minimize the Chamfer Distance.

independently, and build uncertainty estimates over point clouds, produced by densely touching the objects surface with point-based touch sensors [7, 73, 28, 18, 46]. These methods do not make use of learned object priors, and a large number of touches sampled on an object's surface (over 100) is necessary to produce not only a prediction of the surface but also to drive exploration. However, accurate estimates of object shape have also been successfully produced with orders of magnitude fewer touch signals, by making use of high resolution tactile sensors such as [75], large datasets of *static* 3D shape data, and deep learning – see *e.g.* [56, 68]. Note that no prior work exists to learn touch exploration by leveraging large scale datasets. Moreover, no prior work explores active touch in the presence of visual inputs either (*e.g.* an RGB camera).

Combining the recent emergence of both data-driven reconstruction models from vision and touch systems [56, 68], and data-driven active vision approaches, we present a novel formulation for active touch exploration. Our formulation is designed to easily enable the use of vision signals to guide the touch exploration. First, we define a new problem setting over active touch for 3D shape reconstruction where touch exploration strategies can be learned over shape predictions from a learned reconstruction model with strong object priors. Second, we develop a simulator which allows for fast and realistic grasping of objects, and for extracting both vision and touch signals using a robotic hand augmented with high-resolution tactile sensors [75]. Third, we present a 3D reconstruction model from vision and touch which produces mesh-based predictions and achieves impressive performance in the single view image setting, both with and without the presence of touch signals. Fourth, we combine the simulator and reconstruction model to produce a tactile active sensing environment for training and evaluating touch exploration policies. The outline for this environment can be viewed in Figure 1.

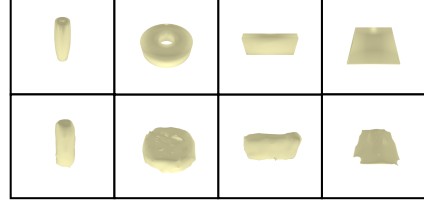

(a) Touch only exploration.

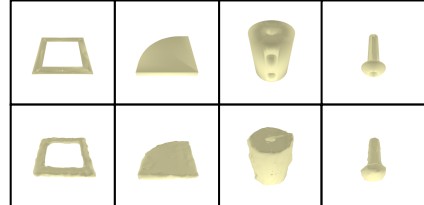

(b) Touch exploration with visual prior.

Figure 2: Target objects (top rows) and predicted 3D shapes (bottom rows) after 5 grasps have been selected.

Over the provided environment, we present a series of data-driven touch exploration models that take as an input a mesh-based shape reconstruction and decide the position of the next touch. By leveraging a large scale dataset of over 25k CAD models from the ABC dataset [34] together with our environment, the data-driven touch exploration models outperform baseline policies which fail to leverage learned patterns between object shape or the distribution of object shapes and optimal actions. We demonstrate our proposed data-driven solutions perform up to 18 % better than random baselines and lead to impressive object reconstructions relative to their input modality, such as those demonstrated in Figure 2. Our framework, training and evaluation setup, and trained models are publicly available on a GitHub repository to ensure and encourage reproducible experimental comparison [3].

---

## 2 Related Work

**3D reconstruction.** 3D shape reconstruction is a popular and well studied area of research with a multitude of approaches proposed across varying 3D representations [13, 63, 19, 52, 67, 23, 12, 71] and input modalities [27, 53, 69, 57]. While still a niche topic, a number of works have attempted to reconstruct 3D shape from only touch signals [7, 49, 59, 45, 38]. These works all assume point based touch signals which supply at most one point of contact of normal information. A limited number of methods have also been proposed for 3D reconstruction from both vision and touch signals [9, 26, 22, 68, 69, 56]. In contrast to these works which use static datasets, we focus on improving 3D reconstruction through tactile exploration. Shape from interaction methods have also been proposed for object reconstruction through hand interactions [42, 64, 47], though in contrast to our work an image of the interaction provides the additional information rather than touch readings.

**Active sensing for reconstruction.** Active vision aims to manipulate the viewpoint of a camera in order to maximize the information for a particular task [11, 76]. Similar to our setting, active vision can be useful in 3D shape reconstruction [62, 70, 17, 41, 14, 35], but where camera perspectives are planned instead of grasp locations. Only very recently have deep learning active vision approaches been proposed for 3D object reconstruction [77, 5, 41, 51]. Deep learning based active sensing for reconstruction has also recently emerged in the medical imaging domain, where the time spent performing MRI scans has been reduced by learning to select a small number of more informative frequencies over a pre-trained reconstruction model [29, 78, 50, 6].

**Touch-based active exploration for shape understanding.** Initial works tackling active tactile exploration focused on object recognition. [1–3]. While some of these methods do integrate both vision and touch for active shape understanding, they only focus on object recognition, operate over small datasets of already known and observed objects and do not leverage deep learning tools for learning patterns over large datasets of object shapes. Fleer and Moringen et. al. proposed to learn haptic exploration strategies using a reinforcement learning framework over a recurrent attention model, however these strategies were optimized for object classification and only trained over a dataset of 4 objects with a single, floating, depth-based tactile sensor array [20].

Several priors works have explored the problem of active acquisition of touch signals specifically for for surface reconstruction. A common theme among most approaches is to predict object shape using Gaussian processes so that uncertainty estimates are naturally available to drive the selection of the next point to touch, which is performed using a single finger [28, 73, 18]. To improve the speed of shape estimation during tactile exploration Matsubara et. al. [40] consider both uncertainty and travel cost when selecting touches using graph-based path planning. Bierbaum et al. [8] instead drove exploration through a dynamic potential field approach made popular in robot navigation to produce point cloud predictions, and extracted touch signals using a comparatively unrealistic 5 finger robotic hand model in simulation. These methods all make use of point based tactile sensors and use deterministic strategies, which are tuned and evaluated over a very small number of objects. . In contrast to these works, our approach is fully data driven, and as such, actions are selected using policies trained and evaluated over tens of thousands of objects. Moreover, predictions are made in mesh space fusing visual information with high resolution tactile signals extracted from objects in simulation using a realistic robot hand.

## 3 Active touch exploration

In our proposed active touch exploration problem, given a pre-trained shape reconstruction model over touch and optionally vision signals, the objective is to select the sequence of touch inputs which lead to the highest improvement in reconstruction accuracy. To tackle this problem, we define an active touch environment that contains a *simulator*, a *reconstruction model* (a pre-trained neural network), and a *loss function*. The simulator takes as input a 3D object shape $O$ together with parameters describing a grasp, $g$, and outputs touch readings, $t$, of the 3D shape at the grasp locations along with an RGB image of the object, $I$. The reconstruction model is a neural network parametrized by $\phi$, which takes an input $X$ and produces the current 3D shape estimate $\hat{O}$ as follows: $\hat{O} = f(X; \phi)$. In our setup, we investigate two reconstruction model variants which differ in their inputs: 1) the model only receives a set of touch readings, $t$, such that $X = t$, and 2) the model receives both a set of touch readings, $t$, and an RGB image rendering of the shape, $I$, such that $X = \{t, I\}$. The loss function takes as input the current belief of the object's shape, $\hat{O}$, and the ground truth shape, $O$, and computes the distance between them: $d(O, \hat{O})$. Thus, active touch exploration can be formulated as sequentially selecting the optimal set of $K$ grasp parameters $\{g_1, g_2..., g_K\}$ that maximize the similarity between the ground truth shape $O$ and the

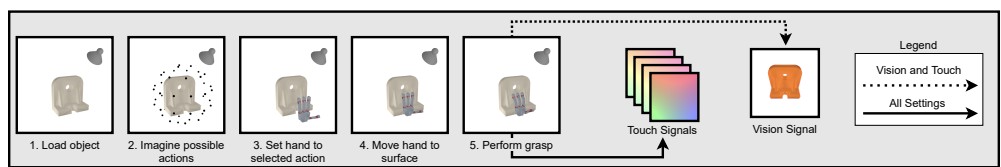

Figure 3: Steps used in the tactile grasping simulator to produce simulated vision and touch signals.

reconstruction output after $K$ grasps $\hat{O}_K$ [4]:

$$\underset{g_1, g_2, \ldots, g_K}{\arg\min} \ d(O, \hat{O_K}), \tag{1}$$

where $g_k$ determine the touch readings fed to the reconstruction network producing $\hat{O}_k$. We use the Chamfer distance (CD) [60] between the predicted and target surface as the distance metric in our active touch formulation. Further details with respect to the CD are provided in the supplemental materials. In the reminder of this section, we provide details about the touch simulator and the reconstruction model.

### 3.1 Active touch simulator

Conceptually, our simulator can be described in the five steps depicted in Figure 3. First, an object is loaded onto the environment. Second, an action space is defined around the 3D object by uniformly placing 50 points on a sphere centered at the center of the loaded object. Third, to choose a grasp, one of the points is selected and a 4-digit robot hand is placed such that its 3rd digit lies on the point and the hand's palm lies tangent to the sphere. Fourth, the hand is moved towards the center of the object until it comes in contact with it. Last, the fingers of the hand are closed until they reach the maximum joint angle, or are halted by the contact with the object. As a result, the simulator produces 4 touch readings(one from each finger of the hand) and one RGB image of the object. Note that each action is defined by its position index on the sphere of 50 actions. This parameterization is selected specifically as it does not require any prior knowledge of the object, other than its center, and in simulation it consistently leads to successful interactions between the hand's touch sensors and the object's surface.

In our simulator, all steps are performed in python across the robotics simulator PyBullet [15], the rendering tool Pyrender [39], and PyTorch [48]. For a given grasp and object, the object is loaded into PyBullet [15], along with a Wonik's Allegro Hand [54] equipped with vision-based touch sensors [36] on each of its fingers, and then the point in space corresponding to the action to be performed is selected and the grasping procedure is performed using PyBullet's physics simulator. Pose information from the produced grasps is then extracted and used by Pyrender to render both a depth map of the object from the perspective of each sensor and an RGB image of the object from a fixed perspective. The depth maps are then converted into simulated touch signals using the method described in [56]. All steps in this procedure are performed in parallel or using GPU accelerated computing, and as a result across the 50 grasping options of 100 randomly chosen objects, simulated grasps and touch signals are produced in $\sim 0.0317$ seconds each on a Tesla V100 GPU with 16 CPU cores. Our simulator supports two modes of tactile exploration *grasping* and *poking*. In the grasping scenario, the hand is performing a full grasp of an object using all four fingers. While, in the poking scenario, only the index finger of the hand is used for touch sensing. Further details on this simulated environment are provided the supplemental materials.

### 3.2 Shape reconstruction

We take a chart-based approach to reconstruction [24], beginning from [56], which used charts to fuse vision and touch signals for shape prediction, and extending it to effectively leverage touch positional information while handling increasing number of touches, and to efficiently predict the object shape from the touch readings In particular, shape is predicted in the mesh representation by repeatedly deforming a collection of independent mesh surface elements – *i.e.* charts – using a graph convolutional network (GCN). The full pipeline for this reconstruction method is highlighted in Figure 4.

**Predicting local shape from touch readings.** Available touch signals are passed through a touch convolutional neural network (CNN), which takes a set of touch readings as input, and produces a set of mesh surface elements, referred to as *touch charts*, representing the surface of the touched object where the touches occurred. We train the touch CNN to directly minimize the CD [60] between the predicted touch charts and the local surface at the corresponding touch site. By contrast, [56] predicts local point clouds which are then converted to charts via iterative optimization.

---

[4]We use a subscript $K$ to indicate that the reconstruction has been obtained after observing $K$ touch readings.

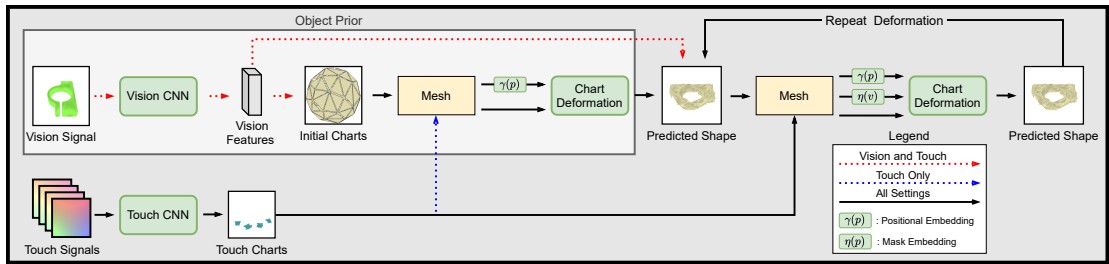

Figure 4: Our pipeline to 3D object reconstruction from vision and touch.

**Chart-based mesh representation and vertex features.** A mesh in the shape of a sphere is initially created from a large collection of mesh surface elements, known as *charts*, where each vertex is defined by its location. We call these mesh surface elements *vision charts* in the presence of vision signals, and *touch charts* in their absence. When leveraging touch signals, the mesh of initial charts is augmented with a set of $N$ additional *touch charts*, which may be initialized with the previous touch CNN vertex location predictions or uninitialized – *i.e.* all vertex locations are set to zero – where $N$ denotes the maximum number of touch charts expected in the active exploration process. Unlike [56], we encode the vertex locations by leveraging positional embeddings [43, 55, 66] to capture higher frequency shape information. When required, a mask embedding is appended to each vertex in the mesh to indicate if it originates from an uninitialized touch chart, a predicted touch chart, or a vision chart. With this setup, a variable number of touch charts can be expected by the subsequent deformation network. In addition, if a vision signal is available, the image is passed through a standard CNN and the extracted vision features are projected onto the vertices of the mesh using perceptual feature pooling [67, 58, 56]. Thus, the resulting mesh is composed of touch and eventually vision charts, whose vertex features include the above-described positional vertex embedding, mask embedding when needed, and if available, the corresponding projected visual features. The mesh connectivity enabling communication among vertices and charts is defined following [56].

**Mesh deformation process.** The chart-based mesh representation is fed to a mesh deformation model composed of two GCNs, by contrast [56] used a single GCN to handle the mesh deformation process. Decoupling the iterative GCN of [56] is crucial to ensure the use of vertex positional information by the model. Since the mesh deformation process of any object starts with the same sphere, parameter sharing across iterations results in the model ignoring the positional vertex information in the predictions of the initial shape belief, hindering the use of the added touch charts in subsequent iterations. In our model, the first GCN aims to learn an object prior, which can be learnt either from vision signals (vision prior) or from touch signals (touch prior). Note that the expected input of the first GCN is static *w.r.t* the position of vision and uninitialized touch charts. The second GCN takes the object belief resulting from the first GCN and refines it through a 2-step deformation process, in which each step recomputes the projection of the image features onto the mesh. Since the input to the second GCN is expected to be different in each case, we can leverage parameter sharing in its 2-step deformation process. Note that touch signals are only included in the second GCN when leveraging vision signals; however, they are included in the first GCN in the touch-only setting when available.

**Training.** The parameters of the whole reconstruction pipeline, including both GCNs and the vision CNN, are jointly optimized to minimize the CD between the predicted and target surface [60, 58]. With this setup, a potential vision signal and a variable number of touch signals can be leveraged to produce a surface prediction in a single model pass. Further details as well as a comparison highlighting the superiority of our shape reconstruction pipeline *w.r.t* [56] can be found in the supplemental material.

## 4 Data-driven policies for touch exploration

Our touch exploration framework takes advantage of the reconstruction model introduced in Section 3.2 to predict 3D shapes in the mesh space, and defines policies to select the positions of the next touches to acquire in order to maximize the similarity between predicted and target shape. While useful for graphics applications and efficient for representing surfaces, meshes are difficult to process and computationally heavy to compare. To combat these issues, we propose to use mesh embeddings of reduced dimensionality to facilitate the learning of our policies. The mesh embedding is extracted from the bottleneck of a mesh autoencoder which is trained offline from the shape predictions to produce a learned embedding space.

We also use the mesh embeddings to allow for efficient distance metric computation over predicted shapes – *i.e.* Euclidean distance in the embedding space.

**Autoencoder for shape embedding.** The encoder, $e$, takes as an input a surface mesh, and produces a mesh embedding. Following our shape reconstruction model, we use positional embeddings [43, 55, 66] to represent the vertices in the mesh. The mesh is then passed through a series Zero-Neighbor GCN layers [58, 32] to update the vertex features, followed by a channel-wise max pooling operation across vertices to produce a latent encoding. The decoder takes the resulting latent encoding and follows the FoldingNet [72] architecture to yield a point cloud with 2,024 points recovering the object shape. The autoencoder is trained by minimizing the CD between the input mesh and the predicted point cloud.

In the remainder of this section, we outline the object-specific and dataset-specific policies selected for the purpose of comparison. In object specific policies the current object shape is considered when deciding which action to perform. In dataset-specific policies the full training set of objects is leveraged to determine the optimal fixed trajectory to perform across all test set objects.

## 4.1 Object-specific policies

**Nearest Neighbor (NN).** We compute *myopic greedy* trajectories for all objects in the training set, and save the object reconstruction at each time step along with the action – grasping parameters – leading to its best immediate improvement. Then, when evaluating, we search our record of training reconstructions and their corresponding actions to find the most similar reconstruction to the current object belief and simply copy the grasping parameters from our record.

To allow for easy and efficient comparisons, we leverage the learned mesh encoder, $e$, to perform similarity search with the $\ell_2$ distance.

**Supervised (Sup.).** In this model we attempt to learn the improvement each action will provide for a given object using regression. An independent network $h_i$ is trained for each time step $i$ to predict the relative improvement which will result from taking each action. Each network is comprised of set of fully connected layers with ReLU activations and takes as input the set of already performed grasp parameters, and the embedding of the current and initial shape reconstruction produced by the pre-trained autoencoder. The networks are trained sequentially, such that the $i$-th network learns to predict the relative improvement of each possible $i$-th action after performing the actions predicted by the networks of the previous time steps. When evaluating the performance of the supervised approach at time step $i$, $h_i$ is used to determine which action will lead to highest improvement, and this action, $g_i$, is selected:

$$g_i = \underset{\mathcal{G} \setminus \{g_0, ..., g_{i-1}\}}{\arg\max} h_i(\{g_0, ..., g_{i-1}\}, e(\hat{O}_0), e(\hat{O}_i)), \tag{2}$$

where $\mathcal{G}$ represents the set of all grasps.

**Double Deep Q-Networks (DDQN).** In the last object-specific model we leverage the discrete deep RL method Double Deep Q-Networks (DDQN) [65]. In our case, the value network takes as input the set of actions already performed and the embedded current reconstruction of the object, and predicts a value for every possible action. We propose two value network architectures. In the first, referred to as $DDQN_m$, the value network takes as input the mesh reconstruction of an object, where an embedding of the performed actions is appended to every vertex's feature vector. The network architecture is identical to that of encoder, $e$, and produces a small shape embedding, which is then passed through a few fully connected layers to predict a value for every action. In the second, referred to as $DDQN_l$, the current reconstruction is passed through the pre-trained mesh encoder, $e$, producing a shape embedding which is concatenated with the embedding of the actions performed, and fed through a few fully connected layers to predict a value for every action. Note that the first setup benefits from a complete understanding of the current shape belief, as the mesh is fed to the value network. The second setup benefits from the pre-computed shape embeddings, which already contain the rich information necessary for reasoning over the object and allows for a simplistic network design which has already been demonstrated to perform well in deep reinforcement learning (RL) settings [65, 21]. In both cases, the action selected is the one which the value network $Q$ predicts has the highest value:

$$g_i = \underset{\mathcal{G} \setminus \{g_0, ..., g_{i-1}\}}{\arg\max} Q(\{g_0, ..., g_{i-1}\}, \hat{O}_i). \tag{3}$$

## 4.2 Dataset-specific policies

**Most frequent best action (MFBA).**

This policy selects the first action by computing the performance of all actions over all objects in the training set, and then chooses the most common best action. For the second action, the performance of performing the first fixed action followed by all remaining actions is computed, and the most common best performing second action is chosen. This is repeated until a full trajectory is obtained. Then, when evaluating, this trajectory is selected every time, regardless of the object reconstruction.

**Lowest error best action (LEBA).** This policy is in effect identical to the MFBA except that the action which leads to the greatest average error improvement is selected and fixed at every time step rather then the most common best action.

## 5 Experiments

In this section, we validate the reconstruction and autoencoder models. We then compare object-specific and dataset-specific policies to several baselines. Additional experimental details can be found in the supplementary material.

### 5.1 Experimental setup

### 5.2 Shape reconstruction from static vision and touch signals

**Dataset of CAD models.** The dataset used is made up of 40,000 objects sampled from the ABC dataset [34, 56], a CAD model dataset of approximately one million objects. This dataset poses a much harder generalization challenge than other 3D object datasets, such as [10] due to its highly variable object shapes, and lack of clearly defined classes over which biases can be learned. The geometry of these objects were decimated such that all objects possess approximately 500 vertices. Those objects which could not be reduced to this size due to geometric constraints and those which possessed multiple

| Model | Grasp # | |
|---|---|---|
| | 0 | 1 |
| $T_G$ [56] | 25.586 $\pm 0.069$ | 9.016 $\pm 0.358$ |
| $T_G$ [ours] | **24.864** $\pm 0.266$ | **8.220** $\pm 0.389$ |
| V&$T_G$ [56] | 2.653 $\pm 0.022$ | 2.637 $\pm 0.042$ |
| V&$T_G$ [ours] | **2.538** $\pm 0.098$ | **2.486** $\pm 0.102$ |

Table 1: Comparison between our reconstruction model and state-of-art in terms of CD.

disconnected parts were automatically removed, leading to set of 26,545 usable object models. This set of objects was split into 5 sets; 3 training sets [5] of size 7,700 object each, a validation set comprised 2,000 objects, and a test set of size 1,000.

**Baselines and Oracle.**

(1) *Random* baseline. As a naive baseline, a random policy is considered, which selects for every time step and object one of the available actions uniformly at random. This is the standard baseline for any exploration algorithm. (2) *Even* baseline. As a second naive baseline, we consider a policy which randomly selects an evenly spaced set of 5 actions over the sphere of possible actions. This baseline is chosen as it will result in uniform coverage of the target object, which is intuitively a useful and strong strategy for object understanding in our task (3) *Oracle*. As a near-optimal target for the performance of our policies a *myopic* oracle policy is considered. In this policy, for a given object and time step the action which resulted in the best improvement is selected.

This policy possesses unfair hindsight, which is not accessible to all others, and so should be seen as an upper-bound point of comparison. The true optimal policy cannot be computed in a reasonable time frame, however due to the diminishing return of rewards from actions in this framework, the provided myopic oracle policy represents a close approximation.

**Experimental scenarios.** We examine the performance of our exploration framework and models across 4 learning settings: (1) *poking only*, $T_P$, where only touch from the third finger of the hand is leveraged; (2) *grasping only*, $T_G$, where only touch signals from all hand sensors are used during shape exploration; (3) *poking with vision*, V&$T_P$, an extension of $T_P$ that includes a visual input signal; and (4) *grasping with vision*, V&$T_G$, an extension of $T_G$ that leverages a visual signal.

We evaluate the performance of our proposed reconstruction method in the target domain of 3D reconstruction from vision and touch, and compare it to the current state-of-the-art [56]. While Wang et. al.

---

[5]Three training sets are used to mitigate the progressive overfitting which might occur from sequentially training the elements of our pipeline (reconstruction, autoencoder and policy networks) on the same data.

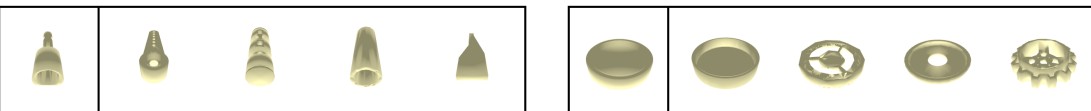

Figure 5: Two objects from the test set, along with their four nearest neighbors in the test set measured in the latent space of our trained autoencoder (V&T$_G$ setting).

| | | **Baselines** | | **Dataset specific** | | | **Object-specific** | | |
| Input | Oracle | Random | Even | MFBA | LEBA | NN | DDQN$_m$ | DDQN$_l$ | Sup |
|---|---|---|---|---|---|---|---|---|---|
| T$_P$ | 19.35 | 36.38 | 33.25 | 32.40 | **29.85** | 33.46 | 32.41 | 31.10 | 31.21 |
| | ±0.00 | ±0.29 | ±0.48 | ±1.04 | ±0.39 | ±0.51 | ±0.40 | ±0.34 | ±0.67 |
| T$_G$ | 16.38 | 25.83 | 24.53 | 23.46 | **23.04** | 24.34 | 23.92 | 23.84 | 23.70 |
| | ±0.00 | ±0.14 | ±0.27 | ±0.07 | ±0.09 | ±0.29 | ±0.14 | ±0.23 | ±0.27 |
| V&T$_P$ | 78.95 | 94.56 | 93.95 | 93.59 | 92.36 | **91.79** | 93.75 | 92.62 | 93.12 |
| | ±0.00 | ±0.34 | ±0.29 | ±0.32 | ±0.25 | ±0.15 | ±0.48 | ±0.30 | ±0.38 |
| V&T$_G$ | 77.18 | 90.65 | 90.29 | 89.39 | 89.31 | **88.53** | 90.07 | 89.32 | 89.46 |
| | ±0.00 | ±0.34 | ±0.32 | ±0.11 | ±0.25 | ±0.25 | ±0.51 | ±0.17 | ±0.23 |

Table 2: Comparison of active touch exploration strategies. Numbers represent a ratio between CD after 5 actions and initial CD (lower is better).

[68] do consider both vision and touch for 3D reconstruction, touch is not fused directly for prediction but rather used for shape refinement in sim2real transfer. The results of this experiment can be seen in Table 1 where the 4 highest performing models have been selected from the validation set while doing hyper-parameter search, and mean and variance numbers across these 4 models on the test set are shown. From these results it is clear that the proposed method outperforms the baseline comparison in all settings, and validates its model choices in our target multi-modal domain.

### 5.3   Shape Autoencoder

For each of the four learning settings an autoencoder is trained by leveraging the output of the reconstruction model. We qualitatively validate the shape embedding learnt by our autoencoder by visualizing object shapes and their nearest neighbors in the learnt embedding space. Figure 5 depicts 2 random objects sampled from the test set, along with the 4 other objects closest to their latent encoding in the test set for the V&T$_G$ setting. Moreover, in the supplemental materials we highlight the average CD between the input and output meshes across 5 grasps, in our 4 learning settings, and observe that they are low relative to the error of the corresponding reconstruction models. The visual similarity of objects to their closest neighbors in the latent space along with the relatively low CD achieved demonstrates that the learned latent encodings possess important shape information which may be leveraged in the proposed active exploration policies.

### 5.4   Active Touch Exploration Results

We examine the performance of all described active touch exploration strategies over 5 grasps and show the results of all policies over the 4 learning settings in Table 2. For the DDQN$_m$, DDQN$_l$, and Supervised policies, the 5 highest performing models have been selected from the validation set while doing hyper-parameter search, and mean and variance numbers across these 5 models on the test set are reported. For the Random, and Even results, the strategies are repeated 5 times over the test set. For the MFBA, LEBA, and NN results, 5 random subsets of 40% of the training data are made to produce different fixed trajectories, or training set latent distributions. Figure 6 highlights the distributions of action selected by each strategy in the T$_G$ and V&T$_G$ settings. Here, the points of all actions on the sphere are transformed into their corresponding UV coordinates in an image, and the intensity value for every pixel corresponding to an action is set to its relative frequency computed over the test set. The visible area of the sphere of actions from the camera's perspective is highlighted in orange, and non-visible in blue. Figure 2 highlights object predictions after 5 grasps with action selected using the DDQN$_l$ model in the T$_G$ and V&T$_G$ settings.

From Table 2, we can see the clear, and expected trend of the Oracle strategy being the best performing and the Random baseline being the worst, followed by the Even baseline which is slightly better. In all cases,

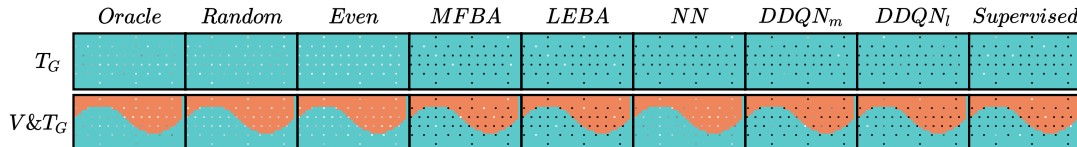

Figure 6: Distribution of selected actions (greyscale encoded) for all policies in the $T_G$ and $V\&T_G$ settings, with visible area of the sphere of actions from the camera highlighted in orange.

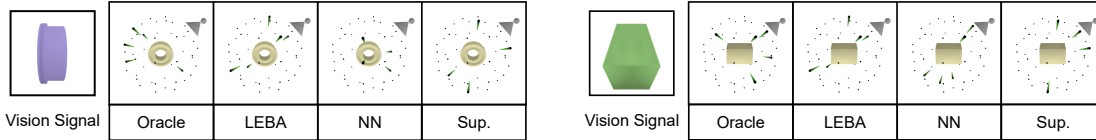

Figure 7: Action selection for the Oracle, LEBA, NN, and Supervised strategies in the $V\&T_G$ setting, where the arrows indicate the direction the hand moves towards the object for each selected action.

LEBA performs better than MFBA, most probably due its selection strategy being directly in line with the target objective. In both touch only settings, the LEBA policy performs best. For the learned policies, even after 4 grasps with a full hand of sensors, enough information about the current shape must not be available to properly learn the best action to perform. This is supported by the distributions in top line of Figure 6 where object-specific polices seem to employ close to a fixed strategy. In contrast to LEBA, these polices cannot leverage the full training dataset simultaneously and so their fixed strategy performs worse. From this we can see that even in the absence of meaningful shape information, leveraging the dataset of objects provides significantly better action selections leading to improved reconstruction accuracy.

In the vision and touch settings ($V\&T_P$ and $V\&T_G$), we see a reverse trend, with the NN policies performing best in both grasping settings. From this we can see that in the presence of better shape understanding (due to additional vision input), more successful action selection can be discriminated per object. This is supported in Figure 6 where we see that with vision priors, far more variable actions are performed, meaning that the object shape is now being appropriately considered. Moreover, in row 2 of Figure 6 a trend can be observed where some policies avoid actions where the touched surface is more likely to be visible from the camera. In the $V\&T_G$ setting, the NN, $DDQN_M$, and supervised policies select visible actions only 40.20%, 45.91% and 43.98% of the time respectively. Compared to the random policy which selects these actions 48.25 % of the time, and the oracle which selects them 43.80% of the time, this indicates that these policies have learned the intuitive strategy of avoiding sampling areas of the surface which have already been observed. In Figure 7 the different action selection strategies employed by various policies and the Oracle are shown for the $V\&T_G$ over 2 randomly sampled objects in the test set. Here the green arrows indicated from which direction the hand moves towards the objects for all 5 action selected. In both cases we see the object specific strategies tend to be fairly uniform around the sides, possibly to correctly identify the unseen dimensions of the objects.

## 6 Discussion

**Limitations.** There exist some limitations which should be highlighted. First, the reconstruction method aims to exclusively minimize the CD, which leads to poor visual object quality in the mesh representation [23, 67]. While attractiveness regularizers are available, we opted to focus exclusively on accuracy to make clear distinctions in improvement from information across different touch options. This is at the cost of visual quality. Second, the chosen shape agnostic grasp parameterization, where the hand always moves towards the center of the object leads touch sites biased towards the center of objects which possess dramatically different dimensional scales. An example of this can be seen the the first object of Figure 7, where because the object is long and thin, all touches will lie in the center of the object, as highlighted by the direction of the arrows. Finally, our environment requires full 3D shape supervision for training, which while easily available in simulation, limits its application to real world scenarios.

**Societal Impact.** Our work contributes towards improved understanding of the three dimensional world in which we all live. In particular, we provide new framework to study touch perception in a simulated environments. Thus, we advance the understanding of the importance of haptic information in the task of active 3D understanding, especially when used in tandem with visual signals. We envision our contributions to be relevant for real world robot-object manipulation. However, the improved active 3D

object understanding could have positive impact in fields beyond robotics, such as automation and virtual reality. Failures of these models could arise if not enough testing is performed prior to the deployment of the automation tools. To mitigate these risks, we encourage further investigation focusing on active 3D reconstruction system generalization limits both in the simulated and real-world scenarios.

**Conclusions.** In this paper, we explored the problem of data-driven active touch for 3D object reconstruction from vision and touch. We introduced a tactile-grasping simulator which allows for the efficient production of vision and touch signals from selected grasp parameters, and built a new 3D reconstruction method from vision and touch which achieves impressive performance both with and without haptic inputs. Over these elements and a large dataset of simulated objects, we built an active touch exploration environment which allows for the training and testing of active touch policies for 3D shape reconstruction. Finally, we produced an array of data-driven active touch policies which we compared to a set of baselines. The benefit of leveraging data for active touch is then highlighted by the superior reconstruction results of learned policies both in the presence of poor and rich shape information. In the presence of only touch information, the most successful exploration strategies learn a deterministic trajectory over the training data to employ indiscriminately over test objects even in the presence of shape information, indicating that either not enough information is present or that this information cannot be learned over with the current models. In the vision and touch settings the most fruitful strategies learn to select grasps based on the current belief of the objects' shape, and experiments also indicate that learned models tend to favour grasps which occur on occluded areas of the objects' surface.

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
