# Appendix

In this appendix, we supply additional information, experiments, and results. This includes a more in depth review of the simulator (Section A), the full definition of the Chamfer Distance (Section B), the **network architecture and training details** (Section C) for all experiments, additional results and experiments (Section D), **licenses** (Section E) for assets and frameworks used, and descriptions of **compute** (Section F) used for all tasks. Sections containing elements from the checklist are boldfaced.

## A  Simulator

In the following section details with respect to the tactile grasping environment will be provided, including the manner in which grasps on an object are performed, and an explanation of how the touch signals are simulated.

### A.1  Grasping

A batch of PyBullet simulator [3] threads are kept constantly available throughout training. Each thread possesses the Allegro hand [11] pre-loaded. When a batch of objects needs to be grasped with a set of chosen actions, each object is loaded into one of the threads such that the object's center is at position $[0, 0, 0.6]$, where the max size of the object across any dimension is $0.5$. In a given thread and with a given action, a fixed sphere of points with radius $1$ is imagined around the object, and the chosen action is used to pick its corresponding point. The hand is then loaded in an open position facing the object with the third digit placed at the point, and the palm of the hand tangent to the imagined sphere. A line from the point to the center of the object is then imagined, and the closest intersection of this line with the convex hull of the object is computed. The hand is moved along the imagined line up to the intersection, and then reoriented so that the palm of the hand is tangent to the surface of the object at the intersection site. The grasp is then performed by increasing all of the hand's joint angles in $5$ maximal joint updates. Here the sensors on the fingers will either touch the object in some way, be impeded from performing the touch by some geometry in the hand or object, or miss the object entirely and the finger will close to the palm. The pose information of the hand for each thread is then exported from the thread.

### A.2  Simulating Touch

A Pyrender [8] scene is kept constantly available throughout training to allow for touch signals to be simulated. The object is first loaded into the scene in the same position and orientation as in the PyBullet thread. For each touch sensor in the hand, the pose information of its finger when the grasp was performed is used to place a simulated perspective camera in its same position and orientation. A depth image is then produced from this camera. This depth image is used to produce a simulated touch signal, through the same method as in [12].

Vision signals from each object are also produced using Pyrender. To improve the simulation time, this is performed offline, and stored images are loaded from memory when needed. To generate this vision signal (image), the object is given a random colour texture and placed alone in an empty scene with $4$ point lights placed in fixed positions around it. The produced images are of size $256 \times 256 \times 3$.

### A.3  Simulation Time

For the complexity of the required simulation task, the time to simulate all 4 touches for a single grasp is relatively quick. This due to a number of optimization choices which include:

- All operations are performed within the same python instance.

- GPU acceleration is leveraged for simulating the touch when possible, such as when rendering with Pyrender, and performing heavy mathematical operations in PyTorch [10].

- The meshes for all objects possess on the order of 500 vertices. As a result, the time required for operations such as loading the mesh into both simulators, performing the physics simulation in PyBullet, and rendering images in Pyrender is minimal.

- The number of PyBullet physics simulation steps required to perform the grasp is only five, and all other computations for the grasps – *e.g.* the hand placement relative to the object – are computed out of simulation.

- Objects remain loaded within Pybullet and Pyrender between grasps in the same trajectory.

As a result of these optimizations, the average time to extract the 4 simulated touch signals from a single grasp across all grasp options and 100 random objects from the third training set is $\sim 0.0317$ seconds on a machine with 1 Tesla V100 GPU and with 16 CPU cores.

# B  Chamfer Distance

The Chamfer distance (CD) is used extensively to compare predictions for training and evaluation in this paper. The Chamfer distance takes as input two set of point clouds and computes the average minimal distance between points in each set. As our prediction are in the mesh representation, we first uniformly sample points on the surface of each mesh, $O$ and $\hat{O}$, to produce point clouds $S$ and $\hat{S}$. We then compute the difference:

$$CD(O, \hat{O}) = \sum_{p_1 \in S} \min_{p_2 \in \hat{S}} \|p_1 - p_2\|_2^2 + \sum_{p_2 \in \hat{S}} \min_{p_1 \in S} \|p_2 - p_1\|_2^2. \tag{1}$$

For all experiments, we sample $30,000$ points from each object, and average over 3 computations of the CD. This is over more points than normally used, and usually no averaging is performed. These changes were employed to address the stochasticity induced by converting our meshes to point clouds, and ensure that changes in CD can be attributed to different touches being performed rather then variance in the approximation.

# C  Network and Training details

In the following section, the network architectures and training details will be described for all learned models.

## C.1  Touch CNN Model

A convolutional neural network (CNN) is trained to convert an input touch signal into a chart representing the local surface of the object where the touch occurred. A chart here is a mesh sheet, meant to represent a small subsection of a full mesh surface. These produced charts are referred to as *touch* charts. To do this, the touch signal is passed though a series of CNN layers, reshaped into a vector, and passed through a series of fully connected layers to predict the vertex positions of a small fix size mesh with 25 vertices. The meshes is then rotated and translated with the known pose information of the finger which performed the touch. The parameters of this network are trained to minimize CD between the produced mesh and the ground truth surface of the object at the location of the touch. The same trained network is used across all active touch settings as the objective of the network does not change between them. The touch signal input image is of size $121 \times 121 \times 3$. The network architecture is provided in Table 5. The network was trained using the Adam optimizer [7] with a learning rate of $0.001$ for 300 epochs at a batch size of 256. Hyper-parameter search was performed in a grid search over learning rate $\{0.0001, 0.001\}$ and batch size $\{128, 256\}$. The performance of the models was evaluated on the validation set every epoch, and the best performing model across these evaluations was selected. The objects used in this training were from the first training set, and in each batch, a set of random touches are sampled from these objects. With this trained network, a touch chart can be produced for any given touch in a grasp, representing local surface where the touch takes place.

## C.2  Reconstruction Model

The model takes as input a set of touch signals and a vision signal, all in the form of RGB images and produces from them a collection of charts which have been deformed and arranged to make up the full surface of the predicted shape. Initially, all touch signals are converted into touch charts using the pre-trained touch CNN.

In the vision and touch setting, a collection of 76 vision charts are formed in the shape of a sphere and the vertices on the borders of these charts share edges with each other to allow communication between them. Recall that the iterative mesh deformation process is performed by two sets of networks which do not share parameters. In the first part of the deformation process, we pass the image signal through a VGG-like CNN with network architecture described in Table 6 to extract image features. We then use perceptual feature pooling [17, 14] to project the extracted image features onto the vertices of our mesh (composed of the above-described charts). The feature vectors this process produces are of size 118. This mesh is then passed through a graph convolutional network with Zero-Neighbor GCN layers [14] with network architecture described in Table 7 and the output of this is added to the original mesh to predict a new location for every vertex. A set of K empty touch charts which are initially given position $(0, 0, 0)$ for all vertices are then appended to the mesh of vision charts, and then every predicted touch chart is used to

replace one of the empty touch charts each. All touch charts have one of its center vertices connected by an edge to every vision chart's border vertices to allow communication between vision and touch charts. A mask embedding of size 118 is produced indicating if each vertex in this graph is from a vision chart, empty touch chart, or predicted touch chart. The vertex positions of each vertex are concatenated with a Nerf positional embedding [9] of length 10 to produce feature vectors which are then passed through 3 fully connected layers with ReLU activations to grow their shape to size 118. In the second part of the deformation process, we again pass the image signal through a VGG-like CNN with network architecture described in Table 6 to extract image features which are projected onto the vertices of the mesh using perceptual feature pooling. The features from the positional embedding, mask embedding and image are then added together and passed again through a graph convolutional network with Zero-Neighbor GCN layers with network architecture described in Table 7. The output of this network is added to the input mesh to predict a new location for every vertex in the vision charts. This process is repeated once more. More precisely, the resulting mesh is passed through the same GCN network to produce a final update to the positions of all vertex charts. The combination of vision and touch charts then makes up the final prediction of shape. The same GCN, CNN, positional embedding, and mask embedding parameters are used in the second and third deformation steps.

In the touch only setting, a collection of 76 vision charts (here referred to as touch charts) is formed in the shape of a sphere and the vertices on the borders of these charts share edges with each other to allow communication between them. An additional set of K empty touch charts, which are initially given position $(0, 0, 0)$ for all vertices, are then appended to the initial mesh of charts. Every predicted touch chart is used to replace one of the empty touch charts in this additional set of charts. All predicted touch charts have one of its center vertices connected by an edge to every other chart's border vertices to allow communications. A mask embedding of size 50 is produced indicating if each vertex in this mesh is from a chart in the initial sphere, empty touch chart, or predicted touch chart. The vertex positions of each vertex are concatenated with a Nerf positional embedding [9] of length 10 to produce a feature vectors which are then passed through 3 fully connected layers with ReLU activations to shrink their size to 50. The features from the positional embedding and mask embedding are then added together and passed through a graph convolutional network with Zero-Neighbor GCN layers with network architecture described in Table 7. The output of this network is added to the original mesh to predict a new location for every vertex. The process of computing these features and passing the resulting mesh through a GCN network to produce an update to the positions of all vertex charts is repeated twice more. The combination of charts then makes up the final prediction of shape. The same GCN parameters are used in the second and third deformation steps, and the positional embedding and mask embedding parameters are used in all three deformations.

In all settings, the reconstruction models were trained to minimize the CD between predicted and target meshes using the Adam optimizer [7] with a learning rate of $0.0001$ for $1,000$ epochs at a batch size of 12 and with patience of 70. In the poking setting K = 5, and in the grasping setting K = 20. A grid search was performed over hyper-parameters including the number of GCN layers $\{8, 10, 12, 15\}$, the hidden dimension size in the GCN layers $\{300, 350, 400\}$, and the percentage of vertex features shared in every Zero-Neighbor GCN layer $\{33\%, 35\%\}$. The performance of the models was evaluated on the validation set every epoch, and the best performing model across these evaluations was selected. The objects used in this training were from the first training set, and for each instance in the batch a random number of grasps between 0 and 5 is sampled.

### C.3  Autoencoder Models

Recall that the autoencoder creates latent embeddings of the predicted shapes produced using the trained reconstructed models. The autoencoder takes as input a mesh in the form of a graph, passes it through a GCN with architecture described in Table 7, concatenates the max value from each vertex feature position across all vertices to produce a features vector, and passes it through 4 fully connected layers with ReLU activations to produce a latent embedding of size 200. For the decoder, the latent embedding is passed through a FoldingNet [18] decoder to produce the predicted point cloud of 2024 points. This setup is trained to minimize the CD between the predicted point cloud and the input mesh. In all settings, the models were trained using the Adam optimizer [7] with a learning rate of 0.0001 for 1000 epochs at a batch size of 12 and with patience of 70. A grid search was performed over hyper-parameters including the number of GCN layers $\{8, 10, 12, 15\}$, the hidden dimension size in the GCN layers $\{300, 350, 400\}$, and the percentage of vertex features shared in every Zero-Neighbor GCN layer $\{8\%, 15\%\}$. The performance of the models was evaluated on the validation set every epoch, and the best performing model across these evaluations was selected in each setting. The objects used in this training were from the second training set, and for each instance in the batch a random number of grasps between 0 and 5 is sampled.

## C.4 DDQN Policies

We make use of the DDQN [16], a standard and highly successful deep reinforcement learning algorithm for solving MDPs over discrete action spaces. In this method, the policy is defined by greedily selecting the action which maximizes a learned value function, at every time step. The value function predicts the value of any given action in the current state, where the value is defined as the expected future cumulative reward. The value function is typically implemented as a deep neural network and is trained to minimize the temporal difference error [15] over sampled data from a replay buffer of previously experienced trajectories in the environment. A full explanation of this method can be found in [16]. In the $DDQN_m$ setting, the action values are predicted directly from the predicted mesh shape. Here, the set of actions performed is described as a $k$-hot mask mask which is passed through 3 fully connected layers with ReLU activations to produce an action embedding of size $100$. A mask embedding of size $100$ is produced indicating if each vertex in the mesh is from a vision chart (charts in the initial sphere), empty touch chart, or predicted touch chart. The vertex positions of each vertex are concatenated with a Nerf positional embedding [9] of length $10$ to produce feature vectors which are then passed through 3 fully connected layers with ReLU activations to grow their shape to size $100$. The action, positional and mask embeddings are then concatenated together and passed though a GCN with network architecture described in Table 7 to produce a feature vector of size $50$ at every vertex. The max value across all vertices in then computed to produce a vector of size $50$ representing the value of each action. In this setting, the models were trained using the standard DDQN framework as described in [16] using the Adam optimizer [7] with a learning rate of $0.001$ for $500,000$ episodes with a network update batch size of $128$. Validation was regularly performed over the validation set to identify the best models. A grid search was performed over hyper-parameters epsilon decay $\{0.999993, 0.999996\}$, discount factor $\{0.9, .99\}$, hidden GCN dimension size $\{100, 200\}$, memory capacity $\{300,000, 100,000\}$, and normalization of the reward by the CD of the initial object belief or the current.

For $DDQN_l$, the action value is predicted over the latent embedding of the predicted mesh. Here, the set of actions performed is again described as a $k$-hot mask passed through 3 fully connected layers with ReLU activations to produce an action embedding of size $50$. This is concatenated with both the latent embeddings of the current mesh and the initial mesh prediction, and then passed though L fully connected layers with hidden dimension H and ReLU activation to produce a vector of size $50$ representing the value of every action. In this setting, the models were trained using the standard DDQN framework as described by [16] using the Adam optimizer [7] with a learning rate of $0.001$ for 500k episodes with a network update batch size of $12$. Validation was constantly performed over the validation set to identify the best models. A grid search was performed over hyper-parameters epsilon decay $\{0.999993, 0.999996\}$, discount factor $\{0.9, .99\}$, number of hidden dimensions H $\{35, 75, 100, 200\}$, number of hidden layers L $\{2, 3\}$, memory capacity $\{300,000, 100,000\}$, and normalization by the CD of the initial object belief or the current.

## C.5 Supervised Policy

For the supervised learning policies, an individual network is trained to learn the value of actions at each of the 5 time steps. The first network is trained to predict the improvement induced by performing every action based on the current belief of the object given that no actions have been performed yet. Here, the improvement of an action is defined as $\frac{CD(\hat{O}_{k+1}, O)}{CD(\hat{O}_k, O)}$, where $k = 0$. Once this network is trained, we move to training network for the second step. Here, for every object in a batch, action one is first performed based on selecting the action with the the highest predicted value from the first network. the second network is trained to predict the improvement induced by performing every action based on the current belief of the object given that one action has been selected from the first network and performed. This continues until all 5 networks have been trained. When testing this policy the action at time step $t$ is selected by passing the current object belief through the $t$-th network and taking the action which value is maximized. In all networks, the set of actions performed is described as a $k$-hot mask which is passed through 2 fully connected layers with ReLU activations and hidden dimension of $100$ to produce an action embedding of size $50$. This is concatenated with both the latent embedding of the current mesh and the initial mesh prediction and then passed though L fully connected layers with hidden dimension H and ReLU activations to produce a vector of size $50$ representing the value of every action. Each network is trained to minimize the mean squared error between the predicted action improvements and the true improvements using the Adam optimizer [7] for $300$ epochs with patience $20$ and batch size $64$. The performance of the models was evaluated on the validation set every epoch, and the best performing model across these evaluations was selected in each setting. A grid search was performed over hyper-parameters:

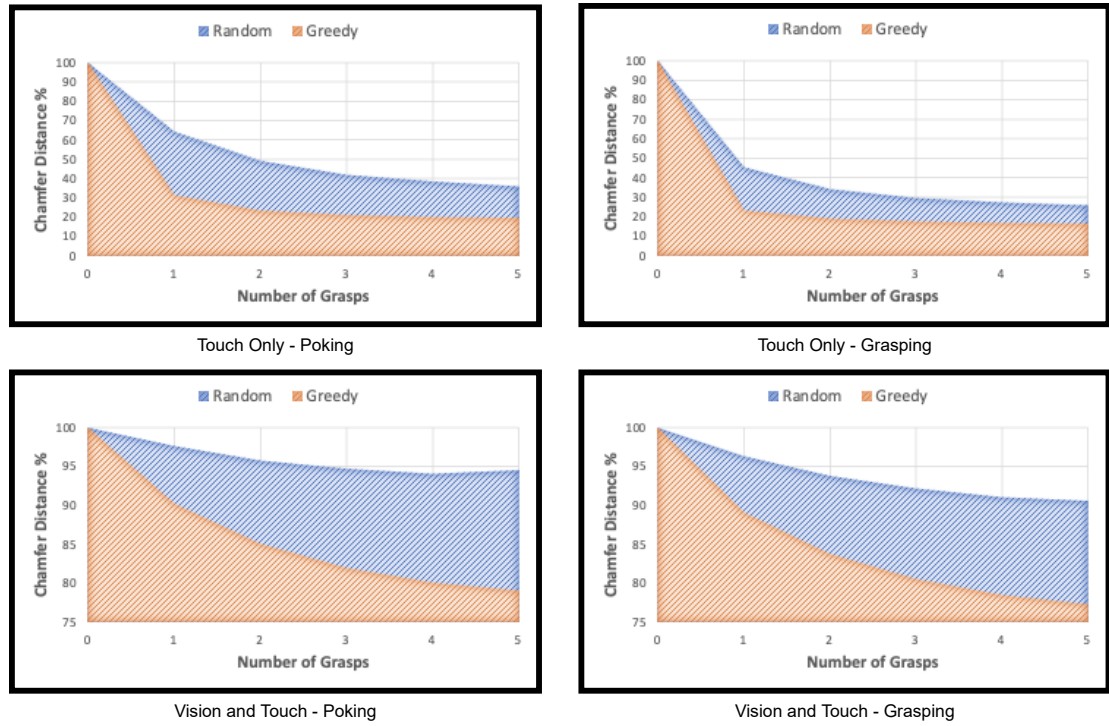

Figure 1: Graphs demonstrating the average reconstruction accuracy of the trained model in each of the 4 learning settings, across different number of grasps and the greedy oracle and random baseline

number of fully connected layers L $\{2, 3, 4\}$, number of hidden dimensions H $\{50, 100, 200\}$ and learning rate $\{0.001, 0.0003\}$.

# D    Additional Results

In this section, additional experiments and results are provided.

## D.1    Mesh Reconstruction

The performance of the best trained reconstruction models on the test set when randomly picking actions across the 4 active touch settings is shown in Table 1. The performance of the best trained models on the test set when greedily picking the best action across the 4 active touch settings is shown in Table 2. Figure 1 shows depicts the change in relative reconstruction accuracy from 0 to 5 touches when picking actions using the greedy and random policies. The large difference in reconstruction performance between the random and greedy policies highlights the need for learned policies which select more informative grasps.

We compare the performance of the proposed model to other state of the art single image 3D object reconstruction models on the 3D Warehouse Dataset [1] using the exact training and evaluation setup described in [5]. The results of this experiment can be seen in Table 3. Here, $F1^{k*\tau}$ is the harmonic mean of the percentage of predicted points with distance at most $k * \tau$ from any ground truth points and the percentage of ground truth points with distance at most $k * \tau$ from any predicted point. The proposed method nearly matches the performance of the best performing method, Mesh-RCNN [5], and notably performs significantly better than the only previous method built for leveraging vision and touch [12].

## D.2    Autoencoder

In Table 4, the chosen autoencoder models' reconstruction Chamfer distances on the test set across all 4 settings are shown. In Figure 2, two random objects are shown in each learning setting along with the 4 closest objects to them in the respective learned latent space of objects. The visual similarity of objects to their closest neighbors in the latent space along with the relatively low CD achieved demonstrates that the learned latent encodings possess important shape information which may be leveraged in the proposed active exploration policies.

| Touches | 0 | 1 | 2 | 3 | 4 | 5 |
|---|---|---|---|---|---|---|
| $T_P$ | 100 | 64.28 | 49.01 | 41.95 | 38.31 | 35.91 |
| $T_G$ | 100 | 45.26 | 34.10 | 29.55 | 27.31 | 25.76 |
| $V\&T_P$ | 100 | 97.66 | 95.76 | 94.71 | 94.09 | 94.51 |
| $V\&T_G$ | 100 | 96.37 | 93.78 | 92.14 | 91.06 | 90.60 |

Table 1: Mesh reconstruction results across all 4 learning settings with actions chosen using the random policy. The units displayed are the percentage of Chamfer distance relative to the Chamfer distance of the initial object belief.

| Touches | 0 | 1 | 2 | 3 | 4 | 5 |
|---|---|---|---|---|---|---|
| $T_P$ | 100 | 31.40 | 23.45 | 20.87 | 19.87 | 19.35 |
| $T_G$ | 100 | 23.09 | 18.99 | 17.49 | 16.76 | 16.38 |
| $V\&T_P$ | 100 | 90.19 | 85.00 | 81.90 | 80.03 | 78.95 |
| $V\&T_G$ | 100 | 89.03 | 83.71 | 8 0.51 | 78.46 | 77.18 |

Table 2: Mesh reconstruction results across all 4 learning settings with actions chosen using the greedy policy. The units displayed are the percentage of Chamfer distance relative to the Chamfer distance of the initial object belief.

| | CD($\downarrow$) | $F1^\tau$ ($\uparrow$) | $F1^{2\tau}$ ($\uparrow$) |
|---|---|---|---|
| N3MR [6] | 2.629 | 3.80 | 47.72 |
| 3D-R2N2 [2] | 1.445 | 39.01 | 54.62 |
| PSG [4] | 0.593 | 48.58 | 69.78 |
| MVD [13] | - | 66.39 | - |
| GEOMetrics [14] | - | 67.37 | - |
| Pixel2Mesh [17] | 0.463 | 67.89 | 79.88 |
| MeshRCNN [5] (Pretty) | 0.391 | 69.83 | 81.76 |
| VT3D [12] | 0.369 | 69.52 | 82.33 |
| MeshRCNN [5] (Best) | 0.306 | 74.84 | 85.75 |
| Ours | 0.346 | 73.58 | 84.78 |

Table 3: Single image 3D shape reconstructing results on the 3D Warehouse Dataset using the evaluation from [5] and [17].

| Grasps | 0 | 1 | 2 | 3 | 4 | 5 |
|---|---|---|---|---|---|---|
| $T_P$ | 0.334 | 0.435 | 0.436 | 0.435 | 0.438 | 0.445 |
| $T_G$ | 0.405 | 0.514 | 0.488 | 0.470 | 0.462 | 0.459 |
| $V\&T_P$ | 0.516 | 0.516 | 0.516 | 0.516 | 0.517 | 0.517 |
| $V\&T_G$ | 0.477 | 0.477 | 0.477 | 0.477 | 0.477 | 0.477 |

Table 4: Autoencoder average Chamfer distance scores across the 4 active learning settings and 5 grasps.

### D.3 Policies

Figure 3 highlights the distributions of action selected by each strategy. Here, the points of all actions on the sphere are transformed into their corresponding UV coordinates in an image, and the intensity value for every pixel corresponding to an action is set to its relative frequency computed over the test set. The visible area of the sphere of actions from the camera's perspective is highlighted in orange, and non-visible in blue. Figure 4 displays shape reconstructions after 5 grasps resulting from the $DDQN_1$ policy. In Figure 5, the different action selection strategies employed by various policies and the Oracle are shown for 2 randomly sampled objects in the test set.

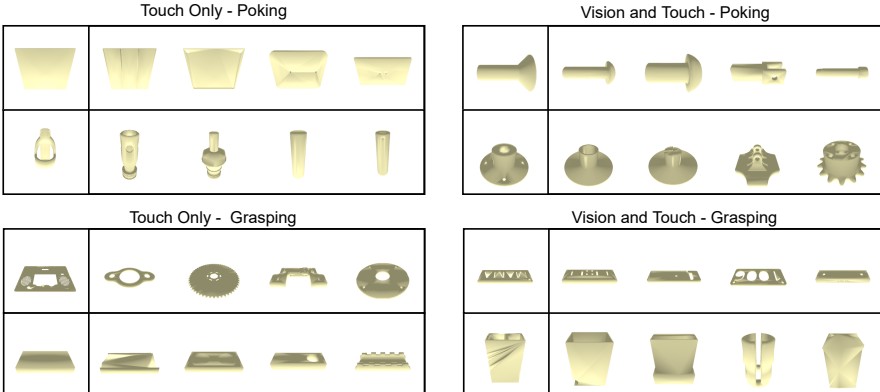

Figure 2: Objects from the test set, along with their four nearest neighbors in the test set measured in the latent space of our trained autoencoder in the 4 learning settings.

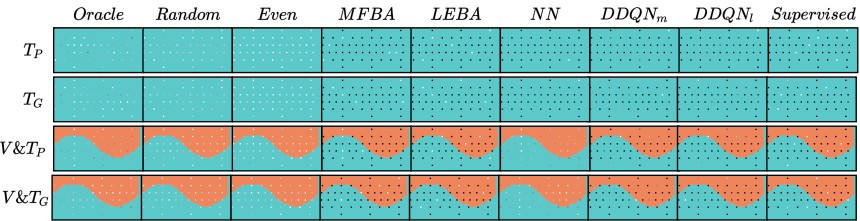

Figure 3: Distribution of selected actions (greyscale encoded) for all policies in all settings, with visible area of the sphere of actions from the camera highlighted in orange.

## E  Licences

All licensed software and assets, along with their licenses are:

1. **PyBullet**: MIT License
   `https://github.com/bulletphysics/bullet3`

2. **PyRender**: MIT License
   `https://github.com/mmatl/pyrender`

3. **Pytorch3D**: BSD 3-Clause License
   `https://github.com/facebookresearch/pytorch3d`

4. **ABC Dataset**: MIT License `https://github.com/deep-geometry/abc-dataset`

5. **Wonik Allegro Hand**: BSD 2-Clause "Simplified" License
   `https://github.com/simlabrobotics/allegro_hand_ros_catkin`

## F  Compute

All described actions were performed on machines with either 1 or 2 Tesla V100 GPUs and with 16 CPU cores.

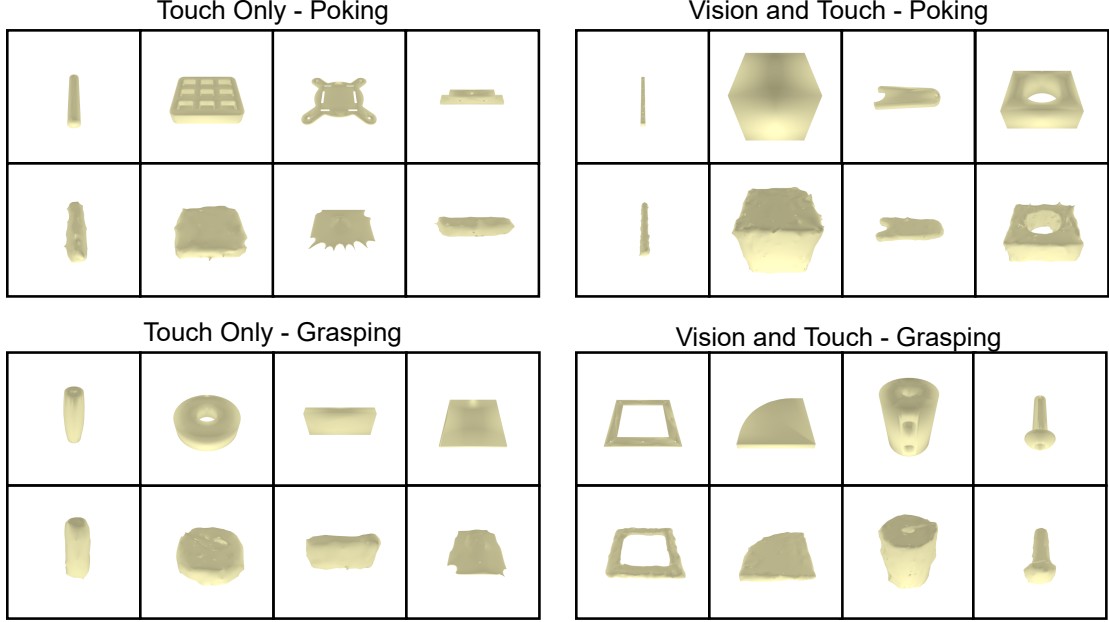

Figure 4: Target objects (top rows) and predicted 3D shapes (bottom rows) after 5 grasps have been selected following the DDQN$_l$ policy in all settings.

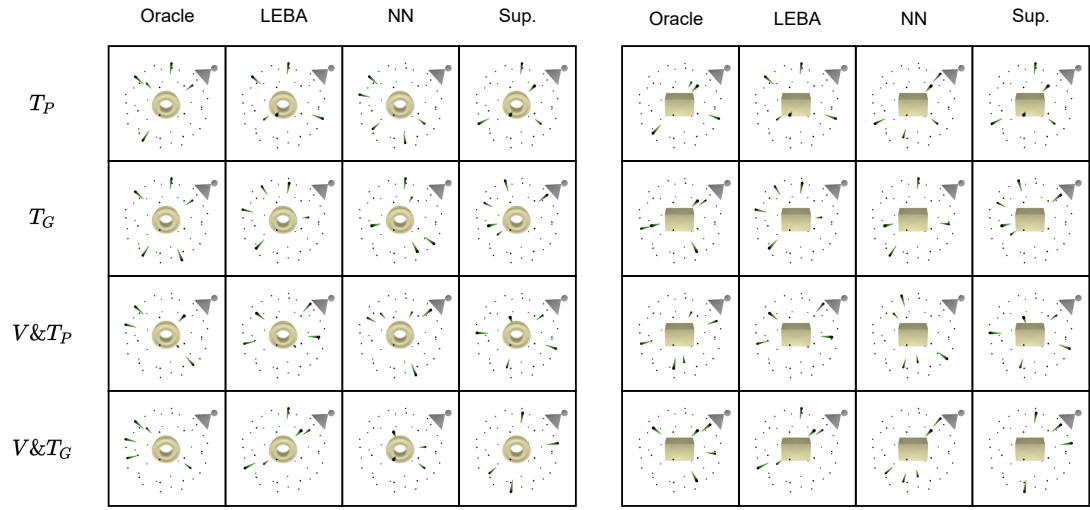

Figure 5: Action selection for the Oracle, LEBA, NN, and Supervised strategies, where the arrows indicate the direction the hand moves towards the object for each selected action.

| Index | Input | Operation | Output Shape |
|---|---|---|---|
| (1) | Input | Conv $(5 \times 5)$ + BN + ReLU | $8 \times 61 \times 61$ |
| (2) | (1) | Conv $(5 \times 5)$ + BN + ReLU | $8 \times 61 \times 61$ |
| (3) | (2) | Conv $(5 \times 5)$ + BN + ReLU | $8 \times 61 \times 61$ |
| (4) | (3) | Conv $(5 \times 5)$ + BN + ReLU | $8 \times 61 \times 61$ |
| (5) | (4) | Conv $(5 \times 5)$ + BN + ReLU | $16 \times 31 \times 31$ |
| (6) | (5) | Conv $(5 \times 5)$ + BN + ReLU | $16 \times 31 \times 31$ |
| (7) | (6) | Conv $(5 \times 5)$ + BN + ReLU | $16 \times 31 \times 31$ |
| (8) | (7) | Conv $(5 \times 5)$ + BN + ReLU | $16 \times 31 \times 31$ |
| (9) | (8) | Conv $(5 \times 5)$ + BN + ReLU | $32 \times 16 \times 16$ |
| (10) | (8) | Conv $(5 \times 5)$ + BN + ReLU | $32 \times 16 \times 16$ |
| (11) | (10) | Conv $(5 \times 5)$ + BN + ReLU | $32 \times 16 \times 16$ |
| (12) | (11) | Conv $(5 \times 5)$ + BN + ReLU | $32 \times 16 \times 16$ |
| (13) | (12) | Conv $(5 \times 5)$ + BN + ReLU | $64 \times 8 \times 8$ |
| (14) | (13) | Conv $(5 \times 5)$ + BN + ReLU | $64 \times 8 \times 8$ |
| (15) | (14) | Conv $(5 \times 5)$ + BN + ReLU | $64 \times 8 \times 8$ |
| (16) | (15) | Conv $(5 \times 5)$ + BN + ReLU | $64 \times 8 \times 8$ |
| (17) | (16) | Conv $(5 \times 5)$ + BN + ReLU | $128 \times 4 \times 4$ |
| (18) | (17) | Conv $(5 \times 5)$ + BN + ReLU | $128 \times 4 \times 4$ |
| (19) | (18) | Conv $(5 \times 5)$ + BN + ReLU | $128 \times 4 \times 4$ |
| (20) | (19) | Conv $(5 \times 5)$ + BN + ReLU | $128 \times 4 \times 4$ |
| (21) | (20) | FC + ReLu | 2048 |
| (22) | (21) | FC + ReLu | 1024 |
| (23) | (22) | FC + ReLu | 512 |
| (24) | (23) | FC + ReLu | 256 |
| (25) | (24) | FC + ReLu | 128 |
| (26) | (25) | FC + ReLu | 75 |
| (27) | (26) | FC | $25 \times 3$ |

Table 5: Architecture for CNN used to convert touch signals into touch charts.

| Index | Input | Operation | Output Shape |
|---|---|---|---|
| (1) | Input | Conv $(5 \times 5)$ + BN + ReLU | $6 \times 256 \times 256$ |
| (2) | (1) | Conv $(5 \times 5)$ + BN + ReLU | $6 \times 254 \times 254$ |
| (3) | (2) | Conv $(5 \times 5)$ + BN + ReLU | $16 \times 126 \times 126$ |
| (4) | (3) | Conv $(5 \times 5)$ + BN + ReLU | $16 \times 124 \times 124$ |
| (5) | (4) | Conv $(5 \times 5)$ + BN + ReLU | $6 \times 122 \times 122$ |
| (6) | (5) | Conv $(5 \times 5)$ + BN + ReLU | $16 \times 120 \times 120$ |
| (7) | (6) | Conv $(5 \times 5)$ + BN + ReLU | $32 \times 59 \times 59$ |
| (8) | (7) | Conv $(5 \times 5)$ + BN + ReLU | $32 \times 57 \times 57$ |
| (9) | (8) | Conv $(5 \times 5)$ + BN + ReLU | $32 \times 55 \times 55$ |
| (10) | (8) | Conv $(5 \times 5)$ + BN + ReLU | $32 \times 53 \times 53$ |
| (11) | (10) | Conv $(5 \times 5)$ + BN + ReLU | $64 \times 26 \times 26$ |
| (12) | (11) | Conv $(5 \times 5)$ + BN + ReLU | $64 \times 24 \times 24$ |
| (13) | (12) | Conv $(5 \times 5)$ + BN + ReLU | $64 \times 22 \times 22$ |
| (14) | (13) | Conv $(5 \times 5)$ + BN + ReLU | $64 \times 20 \times 20$ |

Table 6: Architecture for perceptual feature pooling in vision and touch setting where features from layers 2, 6, 10, and 14 are extracted.

| Index | Input | Operation | Output Shape |
|---|---|---|---|
| (1) | Input | ZN-GCN Layer (C) | $|V| \times H$ |
| (2) | (1) | ZN-GCN Layer(C) | $|V| \times H$ |
| .... | | | |
| K-1 | (K-2) | ZN-GCN Layer (C) | $|V| \times H$ |
| K | (K-1) | GCN Layer | $|V| \times O$ |

Table 7: Architecture for deforming charts positions where H is the chosen hidden dimension size , K is the chosen number of layers, C the percentage of vertex features shared between neighboring vertices in the ZN-GCN layer, and O is the output vertex feature vector size.