# OpenReview forum: "Active 3D Shape Reconstruction from Vision and Touch"
_NeurIPS.cc/2021/Conference — NeurIPS 2021 Poster_

### Official Review · Reviewer_VhZy · 2021-07-13

**Rating:** 6
**Confidence:** 4

**Summary:**

This paper presents a novel active search on touch parameters to improve the 3D shape reconstruction with single (touch) or dual (touch & vision) modalities. The proposed reconstruction network and the active policy show good performance on simulated objects.

**Limitations And Societal Impact:**

Yes

**Main Review:**

1.	On the method originality
This paper mainly has two contributions which are the modified reconstruction network based on [48], and the active search policy of grasp parameters. The whole design is reasonable and novel to me. However, since the modification on the network and the policies are a little direct and intuitive to me, which makes the technical contribution a little limited.
2.	On the experiments
My major concern on the experiments is that they are all experimented in simulation. Since the object is easy to find or 3D printed in the real world, the robot hand (Allegro) is not that expensive (compared to Shadow robot hand), the tactile sensor is open-source for manufacturing with low-cost. And after [48] which has already put forward the discussion of vision-touch learning, I think at this moment, a real-world experiment is needed. I wonder if the simple policy will also work in the real world, the evident challenges would be natural occlusion of object shape, and the displacement after touching.

If the training requires fairly large amount of data, how about apply some simple sim2real strategy? As touch may involves many contact-rich processes, I am suspicious of how well the simulation can mimic the behavior of real touching.

If the pipeline only works with simulation, the applicability will be largely limited.

3.	On the writing
The writing of this paper is quite clear, especially the description about the difference on the reconstruction network. The whole work is easy to read and follow.
4.	On the significance
The integration of vision and touch modalities are important for many applications, especially on robot. 3D shape reconstruction is a key problem when revealing the underlying 3D geometric information from the observations. I think it is a rather under explored area.
5.	Conclusion
Though I have some concerns about the technical contribution and the experiments, I also admit I may set a rather high bar. As the vision-touching learning problem is interesting, important and less explored, I am positive on this work.


**Time Spent Reviewing:**

6

---

> ### Author Response · Authors · 2021-08-09
> **Reply to Reviewer VhZy**
>
> We would like to thank the reviewer for their helpful comments. We were encouraged that the reviewer found the paper easy to follow; the design of the method reasonable and novel; and the problem interesting, important and under-explored. We thank the reviewer for their support of our work. We address the reviewer’s comments below and will include all the feedback in the revised version of the manuscript.
>
> **“This paper mainly has two contributions which are the modified reconstruction network based on [48], and the active search policy of grasp parameters.... the modification on the network and the policies are a little direct and intuitive to me, which makes the technical contribution a little limited”**: Not quite. This is ignoring a main contribution of the paper: the developed simulator and environment for data driven tactile shape exploration. We developed a novel, efficient, and open source simulator for the quick and easy training, and testing of active exploration strategies. The technical contribution here is significant, as it both facilitates this work, and provides an easy tool for future work in this area. As noted by Reviewer Dd2Y : **“The simulator sounds great. Will this also be released publicly?”** Moreover, the modifications of the reconstruction networks are important and allow for significant performance gain. In the touch only setting after one grasp an average percentage improvement of 8.83 % is observed and in the vision and touch setting a 5.72 % improvement is observed which both represent a large gain given the context that a single grasp only supplies a small amount of information about an object shape. Data-driven approaches to active tactile exploration have not been proposed before and we supply a set of natural but successful solutions which lead to significant improvements in performance over baselines. In addition, certain solutions do require a substantial level of complexity, such as the DDQN models which were by no means simple either to implement in the problem setting or to tune to a sufficient performance level.
>
> **“My major concern on the experiments is that they are all experimented in simulation … I wonder if the simple policy will also work in the real world. …  I am suspicious of how well the simulation can mimic the behavior of real touching. If the pipeline only works with simulation, the applicability will be largely limited."**: Identifying successful modeling approaches to robotics tasks proves to very often require multiple iterations and validations in simulation before moving to real world environments [A, B, C], highlighting the need for developing simulation environments. Thus, along these lines, in our paper we open source a simulator for 3D active shape exploration to foster future research and using our simulator we highlight the benefits of data-driven efficient exploration strategies through significant performance gains over baseline approaches. Our simulator allows for the quick and easy training, and testing of possible future solutions to active shape exploration to all who wish to tackle active touch exploration, regardless of funding or robot accessibility. While sim2real transfer of data-driven exploration strategies might be the ultimate goal of this line of work, we believe the contributions made are significant and an essential step toward this ambitious, multidisciplinary goal.
>
> [A] Sim-to-Real Transfer in Deep Reinforcement Learning for Robotics: a Survey,  Zhao et. al. , 2020
>
> [B]  DeepRacer: Educational Autonomous Racing Platform for Experimentation with Sim2Real Reinforcement Learning, Balaji et. al., 2019
>
> [C] Continual Reinforcement Learning deployed in Real-life using Policy Distillation and Sim2Real Transfer, Traoré et. al. , 2019

---

### Official Review · Reviewer_1dhq · 2021-07-16

**Rating:** 7
**Confidence:** 3

**Summary:**

This paper introduces and explores the task of active 3D reconstruction using visual and haptic data. Instead of relying on static data to perform reconstruction, the learning process is guided by exploration, from which new data can be accounted for in the reconstruction process by the proposed model. In order to enable this setting/task, the authors develop a simulator that provides both visual and touch signals that are fed to the model, and an environment for training and evaluating policies.

**Limitations And Societal Impact:**

Yes, the authors have addressed the limitations and societal impact of their work.

**Main Review:**

Strengths:
- The authors introduce the task of active 3D reconstruction, which presents a different setting for reconstruction by leveraging exploration rather than static data, and develop a simulator and environment to enable the task.
- The authors extend the previously proposed chart-based reconstruction method to their setting to better leverage positional information, as well as incorporate additional information collected from exploration.
- The authors use 4 learning settings across a number of data-driven policies to evaluate their approach, including an oracle that represents an upper bound on the performance of the method. Experiments demonstrate the usefulness of active touch through the improved performance on 3D shape reconstruction of the learned policies.
- The paper is well-written, especially when providing an overview of all the policies and baselines used in the experiments.

Weaknesses:
- There are a few design decisions that were skimmed over (e.g., How was 50 selected to be the size of the action space? How was N (max # expected touch charts) determined?)
- The authors use mesh representations, in contrast to point cloud representations used by Smith et al. [48], which introduces additional parameterization with needing to learn an autoencoder to reduce the dimensionality of the features used in learning the policies. Since the autoencoder's objective is to minimize the CD between an input mesh and output point cloud, did the authors try directly using point clouds in their pipeline instead of meshes (i.e., the main difference from [48] being the composition of 2 GCNs for the mesh deformation model)?
- While Table 1 demonstrates better reconstruction in terms of CD, the variance is noticeably higher. Is there any intuition for why this may occur?

----------

Thank you to the authors for addressing and clarifying the concerns raised. My rating remains unchanged.

**Time Spent Reviewing:**

4 hours

---

> ### Author Response · Authors · 2021-08-09
> **Reply to Reviewer 1dhq**
>
> We would like to thank the reviewer for their helpful comments. We are encouraged that the reviewer found the paper well written, and we thank the reviewer for their support of our work. We address the reviewer’s comments below and will include all the feedback in the revised version of the manuscript.
>
> **“How was 50 selected to be the size of the action space? How was N (max # expected touch charts) determined?”**: An extensive parameter search was not performed for either decision. 50 was chosen as it provides enough options to properly cover the surface of the object, and experimentally it was small enough to efficiently explore the search space. 5 was chosen as in all settings it allows for significant change to the shape prediction over the learned reconstruction models and for adequate exploration of the objects. We will highlight these points in Section 5. Importantly, these choices are limited to our experiments, and the simulator which we will release can be set to any number of action choices or maximum grasps.
>
> **“did the authors try directly using point clouds in their pipeline instead of meshes”**: The baseline method [48] does not use the point cloud representation, and uses meshes as well when predicting object shape. As no prior work for combining vision and touch signals makes use of the point cloud representation, we did not attempt this.
>
> **“the variance is noticeably higher. Is there any intuition for why this may occur?”**: We do not have any concrete intuition for the higher variance between the performance of models. We do observe that the higher variance between models in the proposed setup exists also on the validation set. One possible explanation is the proposed model is simply more challenging to optimize over, leading to more variable performance after convergence. However, regardless of the higher variance the performance of the proposed model is still significantly higher than the baseline approach.

---

### Official Review · Reviewer_Dd2Y · 2021-07-17

**Rating:** 8
**Confidence:** 5

**Summary:**

This paper presents an approach for active 3D reconstruction of objects based solely on touch, and optionally together with vision. The main idea is active perception -- estimate where to touch/see the object in order to better reconstruct it. The proposed method to solve this involves a neural network that fuses touch and vision signals to estimate 3D shape, and a policy that predicts where to touch next for best reconstruction performance. Data comes from a simulator that produces both touch and image signals (from the ABC dataset).

**Ethical Concerns:**

No.

**Limitations And Societal Impact:**

I appreciated the paper allocating significant space to discuss limitations and broader impacts. Not many of the papers I have seen do this.

**Main Review:**

I would like to begin by listing several positives with this paper:

- It addresses an important problem (3D reconstruction) and tries to do so using a new active perception paradigm.
- The paper is well written and motivated.
- The experiments are extensive and convincing.

I like the ideas in the paper and the proposed approach. In the following, I will focus on comments, questions, and suggestions for improvement.

- It would be beneficial to differentiate between related previous work [48,59] and the proposed work. The main difference is active perception, but this can be made clearer.

- Figure 2 is missing some labels. What are the top and bottom rows?

- Regarding related work, there is some existing literature on the idea of "Shape from interaction" which would have been nice to discuss. In general, the paper could do a better job of crediting previous papers that present similar ideas. Here's a start:

Shape from interaction, Michel, Zabulis, Argyros, 2014

The paper uses charts for reconstruction but does not cite the most relevant paper:

A papier-mâché approach to learning 3d surface generation, Groueix et al. 2018

- Line 133 mentions 4 touch readings, but it isn't immediately obvious why it's 4 (mention 4 fingers of the robot).

- The simulator sounds great. Will this also be released publicly?

- Is the figure 4 legend correct? Red is denoted as vision and touch but takes only vision signals in the top part.

- I generally like the proposed approach and do not have too many complaints.

- Regarding results and experiments:

    - The results in table 1 show only a marginal improvement over [48], could the authors comment on why?
    - I would have liked to see results on real data, but I understand this is a hard problem.

Overall, this is a great paper that I would like to see published. If the authors can propose a plan to fix the issues above in the rebuttal, that would be great.

**Time Spent Reviewing:**

2

---

> ### Author Response · Authors · 2021-08-09
> **Reply to Reviewer Dd2Y**
>
> We would like to thank the reviewer for their strong support of our work and for their insightful suggestions. We are encouraged that the reviewer liked the ideas and the proposed approach, found the problem interesting, the paper well written, and the experiments extensive and convincing. We address the reviewer’s comments below and will include all the feedback in the revised version of the manuscript.
>
> “**It would be beneficial to differentiate between related previous work [48,59] and the proposed work**”: We will add the following sentence to Section 1: *“In contrast to prior work combining vision and touch signals for 3D object reconstruction [48, 59], we not only learn to predict shape but also learn to select the most informative tactile grasps to maximize accuracy of the prediction.”*
>
> **“Figure 2 is missing some labels. What are the top and bottom rows?”**: A description of the two rows is provided in the figure caption: *“Target objects (top rows) and predicted 3D shapes (bottom rows) after 5 grasps have been selected.”*, though we will move this description to each subfigure to make it more readable.
>
> **“existing literature on the idea of "Shape from interaction" which would have been nice to discuss”**: We thank the reviewer for pointing this out. We will add the suggested reference along with other papers learning object shape from interaction to Section 2: *“Shape from interaction methods have also been proposed for object reconstruction through hand interactions [A, B, C], though in contrast to our work an image of the interaction provides the additional information rather than touch readings”* . We will properly cite AtlasNet by updating Section 3.2: *”We take a chart-based approach to reconstruction [D], beginning from [48], which used charts to fuse vision and touch signals for shape prediction, and extending it to effectively leverage touch positional information while handling increasing number of touches, and to efficiently predict the object shape from the touch readings.”*
>
> **“Line 133 mentions 4 touch readings, but it isn't immediately obvious why it's 4”**: Yes 4 is used as it is the number of fingers on the hand. We will clarify this point on line 133: *“As a result, the simulator produces 4 touch readings (one from each finger of the hand) and one RGB image of the object."*
>
> **“The simulator sounds great. Will this also be released publicly”**: Thank you for your enthusiasm. The simulator, object data, and all pretrained models will be released open source and linked to directly in the camera ready paper.
>
> **“Is the figure 4 legend correct?”**: Yes, this is the correct setup. For the vision and touch setting, in the first interaction of the deformation process, only vision signals are used. For the touch only setting, touch signals are used immediately. This is mentioned in lines 193-194. To make this easier to follow we will add a “touch only” figure and “vision and touch” figure to the appendix to clearly explain the flow of information in the two settings.
>
> **“The results in table 1 show only a marginal improvement over [48]”**: While the improvement in reconstruction accuracy may initially appear small, the improvement with respect to the baseline model becomes notable when viewed in terms of percentage improvement. In the touch only setting after one grasp an average percentage improvement of 8.83 % is observed and in the vision and touch setting a 5.72 % improvement is observed.
>
> **“I would have liked to see results on real data”**:  Identifying successful modeling approaches to robotics tasks proves to very often require multiple iterations and validations in simulation before moving to real world environments [E, F, G], highlighting the need for developing simulation environments. Thus, along these lines, in our paper we open source a simulator for 3D active shape exploration to foster future research and using our simulator we highlight the benefits of data-driven efficient exploration strategies through significant performance gains over baseline approaches. Our simulator allows for the quick and easy training, and testing of possible future solutions to active shape exploration to all who wish to tackle active touch exploration, regardless of funding or robot accessibility. While sim2real transfer of data-driven exploration strategies might be the ultimate goal of this line of work, we believe the contributions made are significant and an essential step toward this ambitious, multidisciplinary goal.
>
>
> [A] Shape from interaction, Michel, Zabulis, Argyros, 2014
>
> [B] 3D Object Reconstruction from Hand-Object Interactions , Tzionas, Gall, 2015
>
> [C] Recovering 3D Models of Manipulated Objects through 3D Tracking of Hand-Object Interactions. Panteleris, Kyriazis, Argyros, 2015
>
> [D] AtlasNet: A Papier-Mâché Approach to Learning 3D Surface Generation, Groueix et. al. , 2018
>
> [E] Sim-to-Real Transfer in Deep Reinforcement Learning for Robotics: a Survey ,  Zhao et. al. , 2020
>
> [F]  DeepRacer: Educational Autonomous Racing Platform for Experimentation with Sim2Real Reinforcement Learning, Balaji et. al., 2019
>
> [G] Continual Reinforcement Learning deployed in Real-life using Policy Distillation and Sim2Real Transfer, Traoré et. al. , 2019

---

### Official Review · Reviewer_Ksky · 2021-07-19

**Rating:** 5
**Confidence:** 4

**Summary:**

The paper presents data-driven exploration strategies for 3D reconstruction using visual and tactile data, a haptic simulator and a mesh-based 3D shape reconstruction model.


**Limitations And Societal Impact:**

Yes.

**Main Review:**

The paper addresses an interesting problem. However it can be improved considering the following points. Some of the listed previous works also focus on improving 3D reconstruction through tactile exploration, e.g. choosing where to touch next without exhaustive exploration also demonstrating experiments using real data. Please clarify and highlight the differences of your work in comparison. Another related work that can be included: Active tactile exploration with uncertainty and travel cost for fast shape estimation of unknown objects. The experiments are only conducted in simulation, how would the results be affected in terms of noise and uncertainty which would be increased in case of real data. The difference in performace in Table 1 looks quite small. Why is [48] chosen as a baseline among all related works, please motivate the choice, also motivate the choice of baselines for the exploration experiments. Could you report the value of K (number of touches/grasps) in experiments? Could you summarise the main findings from the experiments based on all the exploration strategies?

**Time Spent Reviewing:**

4

---

> ### Author Response · Authors · 2021-08-09
> **Reply to Reviewer Ksky**
>
> We would like to thank the reviewer for their comments and helpful feedback. We are encouraged that the reviewer finds the problem interesting. We address the reviewer’s comments below and will include all the feedback in the revised version of the manuscript.
>
> “**Please clarify and highlight the differences of your work in comparison**”: Prior work has tackled active exploration by building shape estimates using gaussian processes without any learned object priors, with their natural uncertainty estimates driving exploration with a point-based touch sensor attached to simple robot end effectors [23, 64, 15]. In contrast, we make use of data-driven models for both reconstruction (using deep learning) and active exploration (such as RL policies) with high resolution tactile sensors and a realistic humanoid robotic hand trained over a relatively huge dataset of over 25 thousand objects. This setup allows for accurate shape prediction over as few as 5 grasps, and for those grasps to be chosen based on strong prior object knowledge as opposed to uncertainty estimates built slowly over significantly more tactile interactions (50+).
>
> “**Another related work that can be included: Active tactile exploration with uncertainty and travel cost for fast shape estimation of unknown objects**": We were not aware of the highlighted work [A] and thank the reviewer for the pointer. While this work does optimize tactile exploration choices from a new perspective, similarly to other related work no form of learning over a large dataset is performed and so far more grasps are still required to build shape and uncertainty understanding than our proposed method. We will add the following sentence to Section 2: *“To improve the speed of shape estimation during tactile exploration Matsubara et. al. [A] consider both uncertainty and travel cost when selecting touches using graph-based path planning.”*
>
> “**The experiments are only conducted in simulation**”: Identifying successful modeling approaches to robotics tasks proves to very often require multiple iterations and validations in simulation before moving to real world environments [B, C, D], highlighting the need for developing simulation environments. Thus, along these lines, in our paper we open source a simulator for 3D active shape exploration to foster future research and using our simulator we highlight the benefits of data-driven efficient exploration strategies through significant performance gains over baseline approaches. Our simulator allows for the quick and easy training, and testing of possible future solutions to active shape exploration to all who wish to tackle active touch exploration, regardless of funding or robot accessibility. While sim2real transfer of data-driven exploration strategies might be the ultimate goal of this line of work, we believe the contributions made are significant and an essential step toward this ambitious, multidisciplinary goal. “**how would the results be affected in terms of noise and uncertainty which would be increased in case of real data**”: While we recognize that sensor noise from the camera, robot joints, and touch sensors are a real problem which must be tackled when undertaking transfer to the real world, we do not speculate on its effect as it would be at best an educated guess and out of scope of the current project. That said, introducing sensor noise to the simulator would be possible, though accurate calibration would be required.
>
> “**The difference in performance in Table 1 looks quite small**“: While the improvement in reconstruction accuracy may initially appear small, the improvement with respect to the baseline model becomes notable when viewed in terms of percentage improvement. In the touch only setting after one grasp an average percentage improvement of 8.83 % is observed and in the vision and touch setting a 5.72 % improvement is observed which both represent a large gain given the context that a single grasp only supplies a small amount of information about an object's shape.
>
> “**Why is [48] chosen as a baseline among all related works, please motivate the choice**”:  We will update Section 5.2 as follows: *“We evaluate the performance of our proposed reconstruction method in the target domain of 3D reconstruction from vision and touch, and compare it to the current state-of-the-art [48]. While Wang et. al. [59] do consider both vision and touch for 3D reconstruction, touch is not fused directly for prediction but rather used for shape refinement in sim2real transfer.”* “**motivate the choice of baselines for the exploration experiments**”: Section 5.1 will be updated as follows: *“(1) Random baseline. As a naive baseline, a random policy is considered, which selects for every time step and object one of the available actions uniformly at random. This is the standard baseline for any exploration algorithm. (2) Even baseline. As a second naive baseline, we consider a policy which randomly selects an evenly spaced set of 5 actions over the sphere of possible actions. This baseline is chosen as it will result in uniform coverage of the target object, which is intuitively a useful and strong strategy for object understanding in our task”*.
>
> “**Could you report the value of K**” : We will report the number of grasps, 5, in Section 5.4.
>
> “**Could you summarise the main findings from the experiments based on all the exploration strategies**”: The main findings of the experiments are highlighted in Section 5.4, lines 333-361. However we will we update the conclusion in Section 6  for clarity as follows: *“The benefit of leveraging data for active touch is then highlighted by the superior reconstruction results of learned policies both in the presence of poor and rich shape information. In the presence of only touch information, the most successful exploration strategies learn a deterministic trajectory over the training data to employ indiscriminately over test objects even in the presence of shape information, indicating that either not enough information is present or that this information cannot be learned over with the current models. In the vision and touch settings the most fruitful strategies learn to select grasps based on the current belief of the objects’ shape, and experiments also indicate that learned models tend to favour grasps which occur on occluded areas of the objects’ surface.”*
>
> [A] Active tactile exploration with uncertainty and travel cost for fast shape estimation of unknown objects, Matsubara et. al. 2017
>
> [B] Sim-to-Real Transfer in Deep Reinforcement Learning for Robotics: a Survey, Zhao et. al. , 2020
>
> [C]  DeepRacer: Educational Autonomous Racing Platform for Experimentation with Sim2Real Reinforcement Learning, Balaji et. al, 2019
>
> [D] Continual Reinforcement Learning deployed in Real-life using Policy Distillation and Sim2Real Transfer, Traoré et. al. , 2019

---

> > ### Author Response · Authors · 2021-08-18
> > **Has Our Response Addressed Your Concerns**
> >
> > Hello reviewer Ksky, we would appreciate it if you could confirm if our response has addressed your concerns, and let us know if any issues remain. Thank you!

---

### Decision · Program_Chairs · 2021-09-27

**Decision:**

Accept (Poster)

**Comment:**

The paper proposes a learning-based approach to 3D reconstruction using a combination of (optional) visual and tactile measurements. A modified neural network takes as input visual and tactile observations and outputs an estimate of the object's 3D shape. This network is coupled with an active perception policy that chooses the next contact point to facilitate tactile-based reconstruction. The method is evaluated through a newly proposed simulator that provides synthetic tactile and visual measurements.

The reviewers agree that the idea of combining tactile observations with visual measurements in an active fashion is a compelling approach to object-scale 3D reconstruction. The paper is well written and easy to follow, and provides an extensive set of experiments that demonstrate the effectiveness of the approach. However, the experiments are limited to simulated data and thus it is not clear how well the framework would perform when operating with real-world visual and tactile measurements. The authors are encouraged to provide a more compelling discussion of the generalizability of the method and, ideally, include real-world experimental results.